# Formation of ice particles through nucleation in the mesosphere

Kyoko K. Tanaka [1], Ingrid Mann [2], and Yuki Kimura [3]

[1]Astronomical Institute, Tohoku University, 6-3, Aza Aoba, Aramaki, Aoba-ku, Sendai, 985-8578, Japan
[2]Institute of Physics and Techonlogy, Department of Physics and Technology, University of Tromsø
[3]Institute of Low Temperature Sciences, Hokkaido University, Kita-19, Nishi-8, Kita-ku, Sapporo, 060-0819, Japan

**Correspondence:** Kyoko K. Tanaka (kktanaka@astr.tohoku.ac.jp)

**Abstract.**

Observations of polar mesospheric clouds have revealed the presence of solid ice particles in the upper mesosphere at high latitudes; however, their formation mechanism remains uncertain. In this study, we investigated the formation process of ice particles through nucleation from small amounts of water vapor at low temperatures. Previous studies that used classical nucleation theory have shown that amorphous solid water particles can nucleate homogeneously at conditions that are present in the mesosphere. However, the rate predictions for water in classical nucleation theory disagree with experimental measurements by several orders of magnitude. We adopted a semi-phenomenological model for the nucleation process, which corrects the evaluation of the molecular cluster formation energy using the second virial coefficient, which agrees with both experiments and molecular dynamics simulations. To calculate the nucleation process, we applied atmospheric conditions for the temperature, pressure, numerical density of dust grains, and cooling rate. The results indicate that homogeneous water nucleation is extremely unlikely to occur in the mesosphere, while heterogeneous nucleation occurs effectively. Dust grains generated by meteor ablation can serve as nuclei for heterogeneous nucleation. We also showed that the ice can form directly in a crystalline state, rather than an amorphous state.

## 1 Introduction

The summer polar **mesopause region**, located at altitudes of 80–90 km, is the coldest part of the Earth's atmosphere. Clouds of ice particles can form at such heights, some of which are visible from the ground and are referred to as noctilucent clouds(**Jasse, 1885; Vestine, 1934; Vaste, 1993**). Noctilucent clouds are generally observed before sunrise and after sunset. Under similar conditions and at overlapping heights, strong radar echoes are observed, known as polar mesospheric summer echoes. Noctilucent clouds are related to the presence of water ice particles (Rapp and Lübken, 2004). **Noctilucent clouds have been studied over long time periods, even half a century (Kirkwood and Stebel, 2003). Noctilucent clouds are also known as polar mesospheric clouds. Polar mesospheric clouds have been observed by satellites since the 1970s (Donahue et al., 1972; Hervig et al., 2012; DeLand et al., 2006).** The ice particles observed in noctilucent clouds comprise particles that are typically tens of nanometers in size **(e.g., Thomas and McKay, 1985; von Cossart et al., 1999; Gumbel and Megner, 2009),** which are large enough to scatter light effectively and therefore, can be detected using a variety of optical remote sensing methods. **Long-term satellite observations have shown that the brightness and frequency of PMCs have been increasing with time**

**(Thomas et al., 2003; DeLand and Thomas, 2015). It is suggested that this is because of the rise of $H_2O$ concentration and that noctilucent clouds are long-term indicators for climate change (e.g., Thomas et al., 1989; Lübken et al., 2018).**

During summer, the high-altitude upper mesosphere can reach temperatures of 130 K. Propagating gravity waves disturb the vertical temperature profiles within the mesospheric cloud layer**(Witt, 1962; Dalin et al., 2012)**. The temperature at this altitude is highly variable. **The lowest temperature is close to 100 K (e.g., 100 K for Lübken et al. (2009) and 110 K for Rapp et al. (2002)).** At this low temperature, even a very small amount of water vapor can achieve a supersaturated state, indicating that water vapor can nucleate and particles can grow. The ice particles grow further as they sediment and are transported vertically in the atmosphere (Rapp et al., 2002), and can encounter different ambient temperatures. **Clouds in the troposphere has been considered to be usually created by a heterogeneous nucleation, on meteoric smoke (see e.g., Rapp and Thomas 2006 for a discussion). However homogeneous nucleation has been considered feasible again after Lübken et al. (2009) reported enormous temperature variability due to gravity waves (Zasetsky et al., 2009; Murray and Jensen, 2010).** Thus, there are two possibilities for the ice particle formation. The first is heterogeneous nucleation, which requires sufficient nuclei, such as dust grains, on which the water vapor deposits. The second is homogeneous nucleation, wherein new water nuclei are formed directly from the gas phase if insufficient impurities are present.

Recent observational results support the hypothesis that ice particles in the mesosphere form as a result of heterogeneous nucleation. Satellite measurements of the atmosphere can be explained using ice particles that contain smaller particles, presumably meteoric smoke. **Meteoric smoke particles form as a result of meteoroid ablation at altitudes of 70–110 km. The major meteoric species are Fe, Mg, Si, and Na which exist as layers of atoms between about 80 and 105 km and atomic ions at higher altitudes. Below 85 km the vapor condenses into agglomerates of oxides, hydroxides, and carbonates with radii of 0.1 to 2 nm (Hunten et al., 1980; Megner et al., 2006),** which can subsequently be used for ice particle formation. The meteoric smoke provides deposition nuclei for ice particle formation.

Hervig et al. (2012) considered the measured extinction of sunlight in the atmosphere due to the presence of ice particles that include fractions of meteoric smoke and found that the volume filling factor of meteoric smoke particles inside ice particles ranges from 0.05 % to several percent. From in-situ rocket observations, Antonsen et al. (2017) **inferred** the size distribution of meteoric smoke particles embedded in larger ice particles, which can be described by inverse power laws with exponents of 3.3–3.7. Experimental studies have also shown that heterogeneous nucleation is possible. Duft et al. (2019) measured heterogeneous ice deposition on iron silicate particles, which they considered to be analogous to meteoric smoke. The **meteoric smoke** particles in the mesosphere are involved in atmospheric air circulation. During this process, coagulation growth can occur (Bardeen et al., 2008, 2010; Megner et al., 2008) and can also be influenced by interaction forces, which depend on the charge state (Baptiste et al., 2021). However, the deposition process remains the critical initial step, and its role in comparison with other growth processes remains uncertain.

Theoretical studies have shown that solid water particles can nucleate homogeneously at mesospheric conditions (Zasetsky et al., 2009; Murray and Jensen, 2010). Murray and Jensen (2010)suggested that the direct homogeneous nucleation of amorphous solid water (ASW) from the vapor phase is possible. They presented a parameterization of homogeneous nucleation based on a modified nucleation theory, wherein they adopted the classical nucleation theory from the vapor phase to ASW,

although ASW is considered to be a meta-stable phase. They also showed that homogeneous nucleation competes with heterogeneous nucleation on meteoric smoke particles when the cooling rate is high ($> 0.5$ K h$^{-1}$). While the classical nucleation theory (CNT) is the most widely used model for describing homogeneous nucleation, it is highly uncertain. It is known that rate predictions based on CNT disagree with experimental measurements for many substances. In the case of water, this deviation is a factor of 10–1000 (Dillmann and Meier, 1991).

A variety of theoretical approaches have been used to develop nucleation theory in previous studies. One of the most successful and useful models is the semi-phenomenological (SP) model, which corrects the formation energy evaluation of a cluster in CNT using the second virial coefficient of a vapor (Dillmann and Meier, 1991). The predictions obtained from the SP model agree surprisingly well with the experimental data for water, nonane, and n-alcohols. In the case of water, the experimental nucleation rate was one to three orders of magnitude smaller than that obtained using CNT, while the SP model was in good agreement within one order of magnitude (Dillmann and Meier, 1991).

In addition to laboratory experiments, numerical approaches, including molecular dynamics simulations, are a powerful method for testing the nucleation model, because the molecular kinetics can be analyzed in detail. To test nucleation theories, molecular dynamics simulations of water vapor nucleation have been performed. A comparison of nucleation models indicates that CNT overestimates nucleation rates by a few orders of magnitude, while the SP model exhibits a better performance (Tanaka et al., 2014; Angelil et al., 2015). Direct large molecular dynamics simulations of homogeneous water nucleation (using up to $4 \times 10^6$ molecules) have allowed extremely low and accurate nucleation to be **derived** (Angelil et al., 2015). A comparison with nucleation models also indicates the validity of the SP model. The results obtained by previous studies may change when a modified model is applied to the nucleation process in the atmosphere. Although many studies have addressed the validity of models of nucleation rates at fixed temperatures, few studies have investigated the changes that occur when these models are applied to natural phenomena where the temperature varies over time. Therefore, it is critical to investigate the effect of using a modified model on the nucleation process.

**In this study, we reconsidered the homogeneous and heterogeneous nucleation as first steps in the formation of ice particles in the mesosphere, with the aim to clarify the formation mechanism of noctilucent clouds.** In particular, we used a model for homogeneous nucleation that agrees with experimental and molecular dynamics simulations, and investigated the effects of using different models to clarify how the modified model affected previous results. We calculated a nucleation process in the cooling vapor using the SP model instead of CNT. The nucleation process depends on atmospheric conditions, including atmospheric temperature, pressure, and cooling rate. We described the homogeneous nucleation process of water droplets from water vapor based on the SP model and solved the temporal evolution of homogeneous nucleation throughout the cooling process. We also investigated the competition process between homogeneous and heterogeneous nucleation at various conditions. We investigated the heterogeneous nucleation process by comparing the parameters to the size distribution and amount of meteoric smoke particles reported by recent studies; however, this study does not consider their properties in detail. Therefore, we use the term "dust" in this study, as the results are generally applicable for solid particles. The conditions under which heterogeneous nucleation occurs effectively depend on the amount of dust grains and the cooling rate. Thus, we

compared the derived conditions required for heterogeneous nucleation with previous observations. We also discuss the particle

crystallization process using the crystallization timescale.

## 2   Methods

### 2.1   Homogeneous nucleation rate

We first considered a formation process of ice particles due to homogeneous nucleation. When the partial pressure of the water

vapor is larger than the equilibrium vapor pressure and becomes supersaturated, water molecules aggregate to form clusters. Cluster growth is promoted when the clusters reach and exceed a critical size. The nucleation rate, which is the number of generated critical clusters in a unit time and volume, is expressed in terms of the free energy of cluster formation (Kalikmanov, 2013). According to the nucleation theory, the nucleation rate $J$ is:

$$J = \left[ \sum_{i=1}^{\infty} \frac{1}{R^+(i)n_e(i)} \right]^{-1} \simeq R^+(i_*)n_e(i_*)Z, \tag{1}$$

where $R^+(i)$ is the transition rate from a cluster of $i$ molecules, $i$-mer, to $(i+1)$-mer per unit time, *i.e.,* the accretion rate, $n_e(i)$ is the equilibrium number density of $i$-mer, and $Z$ is the Zeldovich factor. $R^+(i)$ is given by $R^+(i) = \alpha n_1 v_{\text{th}}(4\pi r_1^2 i^{2/3})$, where $\alpha$ is the sticking probability, $v_{\text{th}}$ is the thermal velocity $(=\sqrt{kT/(2\pi m)})$, $n_1$ is the number density of the monomers. $r_1$ is the radius of a monomer $(= (3m/4\pi \rho_{\text{m}})^{1/3})$ where $m$ is the mass of a molecule and $\rho_{\text{m}}$ is the bulk density. The equilibrium size distribution of a cluster is directly related to the free energy of cluster formation, $\Delta G_i$:

$$\frac{\Delta G_i}{kT} = \ln \left( \frac{n_1}{n_e(i)} \right). \tag{2}$$

There exist three models for the formation energies $\Delta G_i$, the classical nucleation theory (CNT), the modified classical nucleation theory (MCNT), and the semi-phenomenological (SP) model (Dillmann and Meier, 1991; Laaksonen et al., 1994). In each model, the free energy, $\Delta G_i$, is expressed as:

$$\frac{\Delta G_i}{kT} = -i \ln S + \eta i^{2/3}, \tag{3}$$

$$\frac{\Delta G_i}{kT} = -(i-1) \ln S + \eta(i^{2/3} - 1), \tag{4}$$

$$\frac{\Delta G_i}{kT} = -(i-1) \ln S + \eta(i^{2/3} - 1) + \xi(i^{1/3} - 1), \tag{5}$$

where $S = P_1/P_{\text{sat}}$ is the supersaturation ratio of monomers using the saturated vapor pressure $P_{\text{sat}}$ and the partial pressure of monomers $P_1$; $\eta$ and $\xi$ are temperature-dependent quantities that can be fixed from the condensed phase surface tension, bulk density and the second virial coefficient (Tanaka et al., 2014). Note that CNT assumes large cluster sizes, it is not expected to

work for small clusters. In addition, its $\Delta G_i$ does not vanish at $i = 1$, i.e., for monomers, while MCNT and SP models satisfy $\Delta G_i = 0$ at $i = 1$. **The size of critical cluster, $i_*$, is determined by** $dn_e/di = 0$**, *i.e.,***

$$i_* = \left( \frac{2\eta}{3 \ln S} \right)^3 \tag{6}$$

**for the CNT and MCNT, and**

$$i_* = \left( \frac{\eta + \sqrt{\eta^2 + 3\xi \ln S}}{3 \ln S} \right)^3 \tag{7}$$

**for the SP model.** For the thermodynamic quantities, including the surface tension and the saturated vapor pressure of water, we used the data of amorphous ice (Murray and Jensen, 2010). $\eta$ is given by

$$\eta = 4\pi r_1^2 \gamma / kT, \tag{8}$$

where $\gamma$ is the surface tension of the condensed phase. It has been suggested that when homogeneous nucleation occurs, the condensate is likely to be amorphous ice or supercooled droplets (Manka et al., 2012; Murray and Jensen, 2010), so the value of amorphous ice (or supercooled droplet) is used in this study. As for the surface tension, we adopt the data of Murphy and Koop (2005) and Murray and Jensen (2010):

$$
\begin{aligned}
P_{\text{sat}} \quad &= \quad \exp\left[54.842763 - 6763.22/T - 4.21\ln T + 0.000367T + \right. \\
&\qquad \left. \tanh\{0.415(T - 218.8)\}\,(53.878 - 1331.22/T - 9.44523 + 0.014025T)\,\right] [\text{Pa}] 
\end{aligned} \tag{9}
$$

$$\gamma \quad = \quad 235.8 \left( \frac{T_{\text{c}} - T}{T_{\text{c}}} \right)^{1.256} \left[ 1 - 0.625 \left( \frac{T_{\text{c}} - T}{T_{\text{c}}} \right) \right] [\text{erg cm}^{-2}], \tag{10}$$

where the critical temperature of water $T_{\text{c}}$=647.15 K. At 100-170 K, which is the temperature range in this study, the difference in the surface tension is small (87-90 erg cm$^{-2}$).

The monomer radius is derived from the material density. We set $\rho_{\text{m}} = 0.93$ gcm$^{-3}$ (Murray and Jensen, 2010). $\xi$ is a non-dimensional parameter that depends on $T$, which was fixed using the second virial coefficient $B_2$. We fixed the parameter $\xi$ as

$$\xi = -\frac{1}{2^{\frac{1}{3}} - 1} \left[ \ln\left( \frac{-B_2 P_{\text{sat}}}{kT} \right) + (2^{\frac{2}{3}} - 1)\eta \right]. \tag{11}$$

and the second virial coefficient B$_2$ [cm$^3$/mol] is defined as:

$$B_2 = 1000 \left( 0.34404 T_*^{-0.5} - 0.758264 T_*^{-0.8} - 24.219 T_*^{-3.35} - 3978.2 T_*^{-8.3} \right), \tag{12}$$

where $T_* = T/100$ (Harvey and Lemmon, 2004). As shown in Fig. 1, both $\eta$ and $\xi$ increase as the temperature decreases. This indicates that the energy barrier for cluster formation increases because of the increase in $\Delta G_i$. In this case, the nucleation rate decreases. At 100–150 K, the value of $\eta$ is approximately 10, and the value of $\xi$ is approximately 50. This indicates that nucleation occurrence is even more difficult than previously thought.

Using the nucleation rate described above, we solved the basic equations governing non-equilibrium condensation, wherein we considered a gaseous system that cools on a characteristic time scale $\tau$ (Yamamoto and Hasegawa, 1977; Tanaka et al., 2002). The cooling time is defined as $\tau^{-1} = (-1/T_0)(dT/dt)$, where $t$ and $T_0$ are the time and initial temperature, respectively. The basic equation describing ice particle growth is given as:

$$\frac{\partial r(t, t')}{\partial t} = \alpha_{\text{s}} n_1(t) v_{\text{th}} \Omega_1, \tag{13}$$

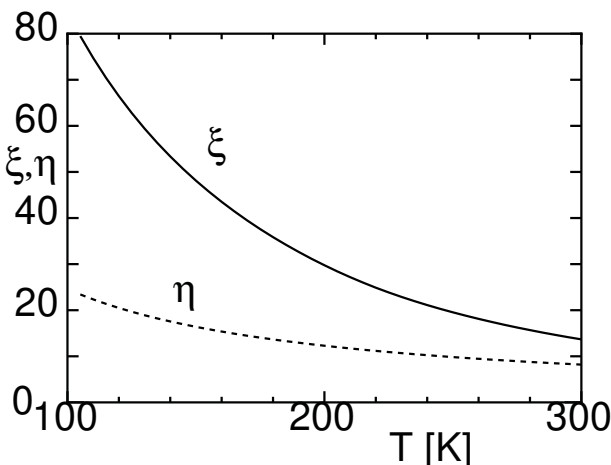

**Figure 1.** Dimensionless parameters $\eta$ and $\xi$ (used to calculate the nucleation rate) and their variations with temperature.

where $r(t,t')$ is the radius formed by homogeneous nucleation at $t$ nucleated at time $t'$, $v_{\text{th}}$ is the thermal velocity of the monomer, and $\Omega_1$ is the monomer volume. The equation describing the consumption of the monomers is as follows:

$$n_1(t) \quad = \quad n_1(0) - \int_0^t J(t') \left( \frac{r(t,t')}{r_1} \right)^3 dt'. \tag{14}$$

**In this study, the initial number density of water molecules is adopted to be the number density at the equilibrium state. As will be discussed in Section 3.1, this value is an upper value since the actual values are determined by a variety of factors.**

## 2.2 Competing process between homogeneous and heterogeneous nucleations

If sufficient dust grains are present in the cloud region, most water molecules will deposit on the surfaces of the dust grains.
However, if the number of dust grains is insufficient, new nuclei form, i.e., homogeneous nucleation occurs. We evaluated the competing process between homogeneous and heterogeneous nucleation and obtained the conditions required for the occurrence of heterogeneous nucleation based on a simple analysis.

In particular, deposition depends on the interfacial energy between the vapor and dust substances. However, meteoric smoke particles are composed of metals and silicates (Rapp and Thomas, 2006; Plane et al., 2015) and water molecules are thought
to deposit quickly on their surfaces (Duft et al., 2019). Therefore, we considered the interfacial energy to be sufficiently small to be negligible. We also assumed that the radii of the dust grains were larger than the critical cluster radius required for homogeneous nucleation, as the vapor will not deposit on the dust grains if their radii are smaller than the critical cluster radius, owing to the effect of the surface energy of water. **As will be shown in Section 3.2**, the radius of the critical cluster is very small, making this assumption reasonable.

Instead of Eq.(14), we used the equation describing the consumption of monomers, given as:

$$n_1(t) = n_1(0) - \int_0^t J(t') \left( \frac{r(t,t')}{r_1} \right)^3 dt'$$

$$- \int_{a_{\text{min}}}^{a_{\text{max}}} A a^{-\lambda} \left( \frac{r_{\text{h}}^3 - a^3}{r_1^3} \right) da, \tag{15}$$

where $r_{\text{h}}$ is the radius of a heterogeneous particle. We considered the dust grain size distribution $n_{\text{d}}(a)$, given by $n_{\text{d}}(a) = A a^{-\lambda}$, with a dust grain radius $a$ and an inverse power exponent $\lambda$, which was set to 2.5 or 3.5 based on observations in this study. The third term on the right-hand side of Eq. (15) corresponds to the monomer consumption, owing to the accretion of the monomer onto the dust grains, and $a_{\text{min}}$ (or $a_{\text{max}}$) is the minimum (maximum) radius of the dust grains. **The number density of the dust grains, $n_{\text{tot}}$, is given by:**

$$n_{\text{tot}} = \int_{a_{\text{min}}}^{a_{\text{max}}} n_{\text{d}}(a) da, \tag{16}$$

**where the constant $A$ in the size distribution is given by**

$$A = \frac{5 n_{\text{tot}}}{2(a_{\text{min}}^{-2.5} - a_{\text{max}}^{-2.5})} \quad \textbf{for} \quad \lambda = 3.5, \quad \textbf{and} \tag{17}$$

$$A = \frac{3 n_{\text{tot}}}{2(a_{\text{min}}^{-1.5} - a_{\text{max}}^{-1.5})} \quad \textbf{for} \quad \lambda = 2.5. \tag{18}$$

We considered the equations describing heterogeneous particle growth, i.e., particle consisting of a dust center and an outer layer of ice, as well as the homogeneous particles given by Eq. (13):

$$\frac{dr_{\text{h}}(t)}{dt} = \alpha_{\text{s}} n_1 v_{\text{th}} \Omega_1, \tag{19}$$

where the initial radius of the heterogeneous particle corresponds to the radius of dust grains $r_{\text{h}}(0) = a$.

We can **roughly** determine whether homogeneous or heterogeneous nucleation is the dominant process based on the fraction of water molecules incorporated into the particle. We considered how much of the water molecule was consumed by heterogeneous nucleation before $t_j$, which is the time of the peak nucleation rate due to homogeneous nucleation. **Here we define a ratio of the number density of monomers which accreted to the particles consisting of a dust center and an outer ice layer formed by the heterogeneous nucleation. We suggest that the condition at which the heterogeneous particle formation starts effectively is:**

$$f = \frac{1}{n_1(0)} \int_{a_{\text{min}}}^{a_{\text{max}}} A a^{-\lambda} \left( \frac{r_{\text{h}}(t_j)^3 - a^3}{r_1^3} \right) da \gtrsim 0.1, \tag{20}$$

**where $f$ is the fraction of water molecules consumed by the heterogeneous particles at $t_j$. Under the assumption that the number density of water molecules at $t_j$ is nearly equal to the initial value $n_1(t_j) \simeq n_1(0)$, the radius of a heterogeneous**

**grain is:**

$$
\begin{aligned}
r_{\mathbf{h}}(t) &\simeq a + \alpha_{\mathbf{s}} n_{\mathbf{1}}(0) v_{\mathbf{th}} \Omega_{\mathbf{1}} t_j \\
&= a + \frac{r_1}{3} \frac{t_j}{\tau_{\mathbf{col}}}.
\end{aligned}
\tag{21}
$$

**Inserting the above equation into Eq. (20), we obtain:**

$$
f = \frac{A}{n_{\mathbf{1}}(0)} \int_{a_{\mathbf{min}}}^{a_{\mathbf{max}}} a^{-\lambda} \left[ \left( \frac{a}{r_1} + \frac{t_j}{3\tau_{\mathbf{col}}} \right)^3 - \left( \frac{a}{r_1} \right)^3 \right] da \gtrsim 0.1,
\tag{22}
$$

**where $\tau_{\mathbf{col}} = (4\pi r_1^2 \alpha_{\mathbf{s}} n_{\mathbf{v}}(0) v_{\mathbf{th}})^{-1}$ is the collision time among monomers. From Eq.(22), we obtain the following condition for $\lambda = 2.5$.**

$$
\frac{A}{n_{\mathbf{1}}(0)} \left[ \frac{6}{r_1^2} \left( \frac{t_j}{3\tau_{\mathbf{col}}} \right) \left( a_{\mathbf{max}}^{0.5} - a_{\mathbf{min}}^{0.5} \right) + \frac{6}{r_1} \left( \frac{t_j}{3\tau_{\mathbf{col}}} \right)^2 \left( a_{\mathbf{min}}^{-0.5} - a_{\mathbf{max}}^{-0.5} \right) \right.
$$
$$
\left. + \frac{2}{3} \left( \frac{t_j}{3\tau_{\mathbf{col}}} \right)^3 \left( a_{\mathbf{min}}^{-1.5} - a_{\mathbf{max}}^{-1.5} \right) \right] \gtrsim 0.1,
\tag{23}
$$

**which yields the conditions for the number density of dust grains when $\lambda = 2.5$:**

$$
n_{\mathbf{tot}} \gtrsim n_{\mathbf{1}}(0) C,
\tag{24}
$$

$$
C = \frac{\left( a_{\mathbf{min}}^{-1.5} - a_{\mathbf{max}}^{-1.5} \right)}{15} \left[ \frac{6}{r_1^2} \left( \frac{t_j}{3\tau_{\mathbf{col}}} \right) \left( a_{\mathbf{max}}^{0.5} - a_{\mathbf{min}}^{0.5} \right) + \frac{6}{r_1} \left( \frac{t_j}{3\tau_{\mathbf{col}}} \right)^2 \left( a_{\mathbf{min}}^{-0.5} - a_{\mathbf{max}}^{-0.5} \right) \right.
$$
$$
\left. + \frac{3}{2} \left( \frac{t_j}{3\tau_{\mathbf{col}}} \right)^3 \left( a_{\mathbf{min}}^{-1.5} - a_{\mathbf{max}}^{-1.5} \right) \right]^{-1} \quad \text{for } \lambda = 2.5.
\tag{25}
$$

**In the same way, we obtained the condition when $\lambda = 3.5$:**

$$
C = \frac{\left( a_{\mathbf{min}}^{-2.5} - a_{\mathbf{min}}^{-2.5} \right)}{25} \left[ \frac{6}{r_1^2} \left( \frac{t_j}{3\tau_{\mathbf{col}}} \right) \left( a_{\mathbf{min}}^{-0.5} - a_{\mathbf{max}}^{-0.5} \right) + \frac{2}{r_1} \left( \frac{t_j}{3\tau_{\mathbf{col}}} \right)^2 \left( a_{\mathbf{min}}^{-1.5} - a_{\mathbf{max}}^{-1.5} \right) \right.
$$
$$
\left. + \frac{2}{5} \left( \frac{t_j}{3\tau_{\mathbf{col}}} \right)^3 \left( a_{\mathbf{min}}^{-2.5} - a_{\mathbf{max}}^{-2.5} \right) \right]^{-1} \quad \text{for } \lambda = 3.5.
\tag{26}
$$

**Although the above condition is obtained from rough estimations, it is useful because it provides a straightforward formulation how the number density of dust particles necessary for the heterogeneous nucleation depends on the dust size and water vapor content.**

## 3  Results

### 3.1  Typical ranges of mesospheric variables

**To obtain the range of parameters that we can assume when investigating the nucleation process, we consider typical values of relevant physical quantities in the region where clouds form in the mesosphere. The Earth's atmosphere at**

this altitude is not in a mean equilibrium state and subject to several influences like for instance the atmospheric transport, chemistry and the solar and magnetospheric effects (Sinnhuber et al., 2012; Sarris, 2019) which are particularly important at high latitudes. Observations are made with lidar, radar and rockets and show that the derived parameters vary spatially and temporarily. Although the mean temperature is between 128 K at the mesopause and 150 K at 82 km (Lübken, 1999), the local minimum temperature is important for the condensation process. The minimum observed temperatures are around 110 K, in some cases as low as 100 K (Lübken et al., 2009) and they are highly variable (Rapp et al., 2002). The concentration of water vapor is considered to be 0.1 to 10 ppmv from observations (Lübken et al., 2009). The concentration of dust grains has been inferred from theoretical considerations and rocket observations in this region. The different estimates of the dust number density range from 1000 to 10000 cm$^{-3}$ (Gumbel and Megner, 2009; Plane et al., 2015; Antonsen et al., 2017). The cooling rate is an important factor to determine the nucleation process. Using a typical gravity wave of amplitude $\sim$10 K and period of a few hours, the cooling rate may be estimated to be few K h$^{-1}$. In the previous study, the cooling rates between about 0.1 to 10 Kh$^{-1}$ were considered (Murray and Jensen, 2010). Bearing in mind the values described above, we make our calculations over a wide range of parameters to investigate the various dependencies. As will be discussed later, the homogeneous nucleation does not occur under mesospheric conditions until the temperature drops to extremely low values. Although the temperatures below 100 K where we find that homogenuous nucleation is important are not realistic for the mesosphere, we here include the results that we obtained at these temperatures for the sake of a discussion.

## 3.2 Homogeneous nucleation

Figure 2 shows a typical example of the homogeneous nucleation at an initial temperature of 135 K, where we solved the basic equations of Eqs. (11) and (12) using the nucleation rate given by Eq. (1). When the initial temperature is 135 K, where the saturation vapor pressure and the number density of water molecules are $2.0 \times 10^{-7}$ Pa and $1.0 \times 10^{8}$ cm$^{-3}$, respectively. In this case, considering an atmospheric pressure of 0.2–0.5 Pa at approximately 85 km, the water vapor fraction corresponds to 0.4–1 ppmv. Similarly, if the temperature is 145 K, the water vapor fraction corresponds to 5–20 ppmv. The observations indicate that there are some variations in the water content, and that the water vapor fraction in the atmosphere is 1–10 ppmv (Berger and vonZahn, 2002; Lübken et al., 2004). Therefore, we considered 135 K and 145 K as typical values in this study. Figure 2 shows the behavior of non-equilibrium condensation of water with a characteristic cooling time of $\tau = 1.5 \times 10^{5}$ s, which corresponds to a cooling rate of $1.0 \times 10^{-3}$ Ks$^{-1}$ (**3.6 Kh$^{-1}$**), in which we used the SP model and assumed the sticking probability of a water molecule to be unity. Because the supersaturation ratio increases exponentially with a decrease in temperature, and because the nucleation rate depends strongly on the supersaturation ratio, the nucleation rate increased sharply. A slight decrease in water molecules due to nucleation caused the nucleation rate to reach its maximum (1 cm$^{-3}$s$^{-1}$) at a temperature $T = T_p$. We call this peak temperature as the nucleation temperature, hereafter. The nucleation temperature $T_p$ was 63 K, and the average radius of the water particles was 4.6 nm. After nucleation, the nucleus grew rapidly, doubling the average radius in $\sim$7 h.

The nucleation temperature and particle size depend on the nucleation model used for calculation. Figure 3 shows the temporal evolution of nucleation rates for the MCNT and SP models. The nucleation temperature was 63 K for the SP model, which was much smaller than the 106 K obtained when using the MCNT model. The average water droplet radii were 4.6 and 1.3 nm for the SP and MCNT models, respectively. A lower nucleation temperature was obtained for the SP model because the free energy for cluster formation $\Delta G_i$ is much larger for the SP model than the MCNT model. **The size of critical nuclei is given by Eqs.(6) and (7).** Due to the high supersaturation ratio, the sizes of the critical clusters are very small in both models, i.e., two and four molecules for the SP and MCNT models, respectively. **For values considered here, the size of critical nuclei ranged from 2 to 10.**

When we performed the calculation using CNT, the nucleation temperature obtained was between those of the MCNT and SP models. For example, the nucleation temperature was 87 K when using CNT, which is between those of the MCNT (106 K) and SP (63 K) models. As noted above, CNT cannot accurately describe $\Delta G_i$ for monomers and has been corrected to be consistent in previous homogeneous nucleation studies. Therefore, we use MCNT instead of CNT for comparison with SP model in this study.

Figure 4 shows the results of the calculations at various cooling rates when the initial temperature was 135 K. For MCNT, the nucleation temperature ranged from $\sim 100$ to 110 K at a cooling rate of $10^{-1}$ **Kh$^{-1}$ to** $10^2$ **Kh$^{-1}$**; however, for the SP model, the nucleation temperature was as low as 100 K. When the cooling rate was 0.36 Kh$^{-1}$, the nucleation temperature was $\sim 80$ K. As the cooling rate increased, the nucleation temperature decreased, reaching 50 K for a cooling rate of $\sim 10$ **Kh$^{-1}$**. Low nucleation temperatures do not match the observations, indicating that homogeneous nucleation is difficult in the mesosphere. In contrast, the size did not change drastically between the two models. For the SP model, the size was larger by a factor of 2. Figure 5 shows the results for an initial temperature of 145 K. For MCNT, the nucleation temperature ranged from $\sim 110$ to 120 K, but for the SP model, it was also as low as 100 K. When the cooling rate was slow ($10^{-1}$ **Kh$^{-1}$**), the nucleation temperature was $\sim 100$ K in the SP model. For initial temperatures of 135 K and 145 K, the initial amounts of water vapor were quite different; the amount of water vapor **in the equilibrium state** was 20 times higher at 145 K than at 135 K. However, the nucleation temperatures were lower for both cases. In contrast, the nucleation temperature was considered to be higher than 100 K based on the observations. To nucleate water homogeneously at a reasonable temperature above 100 K, the cooling rate must be slower than $10^{-2}$ **Kh$^{-1}$ and** $10^{-1}$ **Kh$^{-1}$** for an initial temperature of 135 K or 145 K. **However, the cooling rate such as** $10^{-2}$ **Kh$^{-1}$ is very small and unrealistic for conditions in the mesosphere.**

### 3.3 Condition for heterogeneous nucleation

We investigated the competing process between homogeneous and heterogeneous nucleation and obtained the condition required for the occurrence of heterogeneous nucleation. Figure 6 shows **the number density of dust grains** required for heterogeneous nucleation as a function of cooling time given by Eq. (22) when the initial temperatures **are** 135 K and 145 K and the time at which the homogeneous nucleation rate attains its peak is given by $t_j \simeq \tau$ (Yamamoto and Hasegawa, 1977). In Figs 6 and 7, we adopted $a_{\min} = 0.2$ nm and $a_{\max} = 4$ nm (Baumann et al., 2015). When the amount of dust is large, heterogeneous nucleation occurs. However, when cooling occurs rapidly, homogeneous nucleation is more effective because the

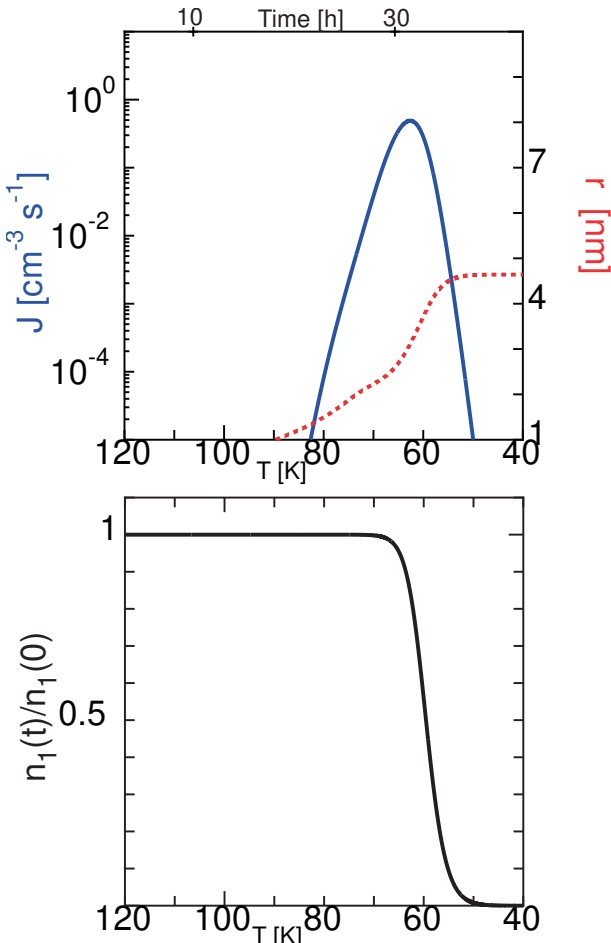

**Figure 2.** Time evolution of the nucleation rate and mean particle radius during homogeneous nucleation **(upper panel), and the ratio of the number density of water molecule to the initial value (bottom panel)** calculated using the SP model. The initial temperature was 135 K and the cooling rate is $3.6\ \mathrm{Kh}^{-1}$.

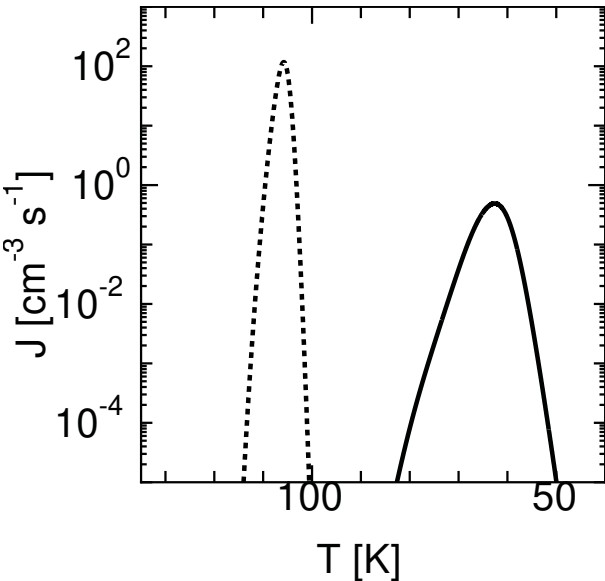

**Figure 3.** Time evolution of the nucleation rates of two models using homogeneous nucleation. The solid curve shows the results of the SP model, while the dotted curve shows the results of the MCNT model.

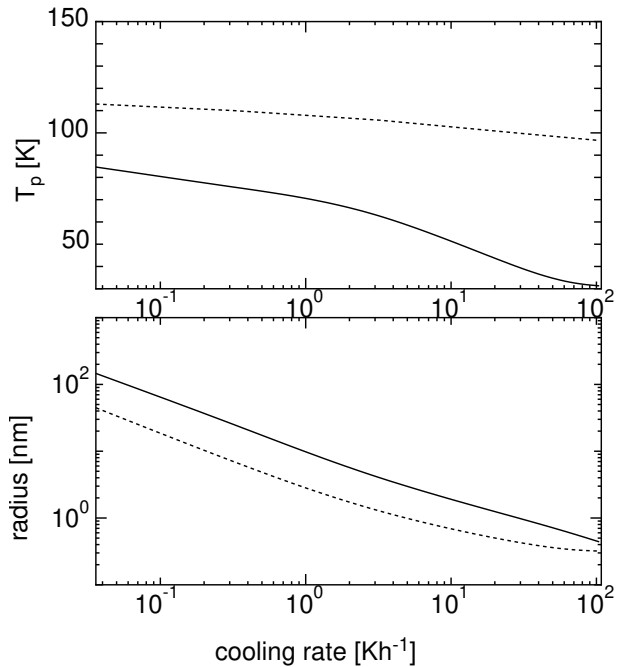

**Figure 4.** Nucleation temperatures and mean particle radii for homogeneous nucleation with an initial temperature of 135 K. Solid curve shows the results of the SP model, dotted curve shows the results of the MCNT model.

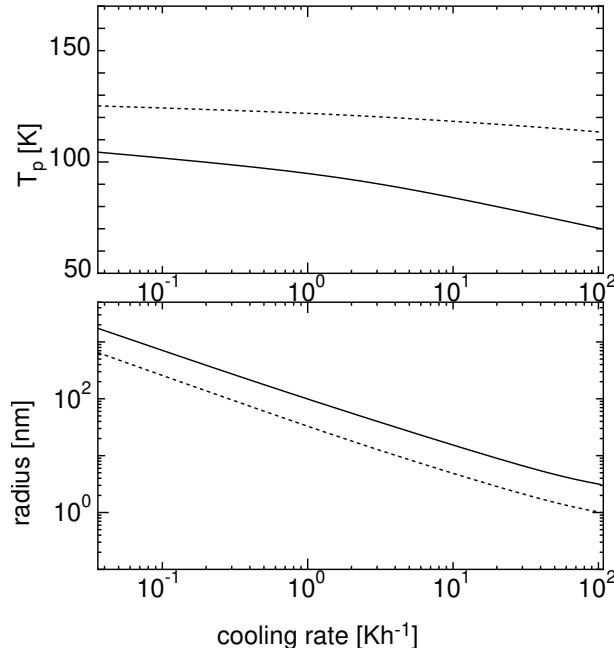

**Figure 5.** The same as Figure 4 but for an initial temperature of 145 K.

supercooling ratio increases quickly. Figures 6 and 7 show a region where **homogeneous nucleation is dominant (where the cooling rate is larger and the dust amount is smaller)**, as well as a possible range in the mesospheric environment where a wide range of cooling rates occur **(0.1 Kh$^{-1}$ to 10 Kh$^{-1}$)**. In Figure 6, the range of number density of the dust grains obtained from the observation is shown (Hervig et al., 2012; Antonsen et al., 2017). From Figures 6 and 7, it is clear that **the observation region is included in the region where heterogeneous nucleation occurs effectively. As can be seen, homogenuous nucleation could occur at cooling rates exceeding roughly 0.1 Kh$^{-1}$; these cooling rates however are typically reached at temperatures below 100 K (see Fig. 4 and Fig. 5) and therefore homogenuous nucleation is not likely.**

### 3.4 Crystallization process

Ice exhibits two potential states when it nucleates in the mesosphere: amorphous or crystalline. However, the state of the ice remains unclear. When water nucleates homogeneously, the first transition is to an amorphous phase with an energetically lower barrier, rather than a stable phase, as described in the Ostwald step rule (Ostwald, 1879). However, experiments on the homogeneous nucleation of water at very low temperatures (∼100 K) have indicated that liquid water or amorphous ice forms (Manka et al., 2012). In contrast, during heterogeneous nucleation, the solid state depends on certain quantities, including pressure and temperature. We introduced a condition for amorphous ice formation based on a simple analysis. This condition was derived by previous studies (Gail and Sedlmayr, 1984; Kouchi et al., 1994), i.e., the diffusion distance of the coverage time of the surface by adatoms is smaller than the lattice constant $a_l(= 4.5 \times 10^{-8}$ cm) of crystalline ice, which yields the following

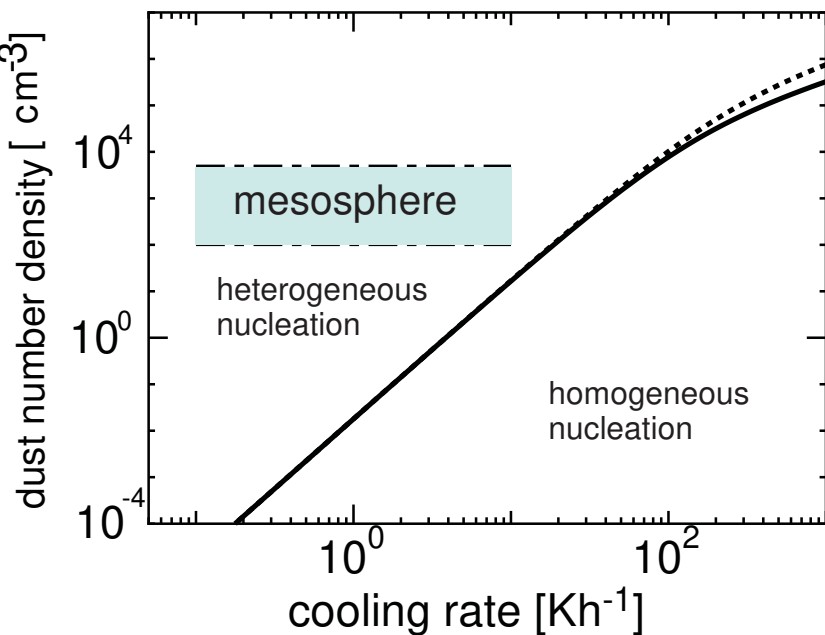

**Figure 6.** The condition of the **number density of dust grains** (vertical axis) and the cooling rate (horizontal axis) required for the hetero-geneous nucleation at an initial temperature of 135 K. Solid and dotted lines represent $\lambda = 2.5$ and $\lambda = 3.5$, respectively.

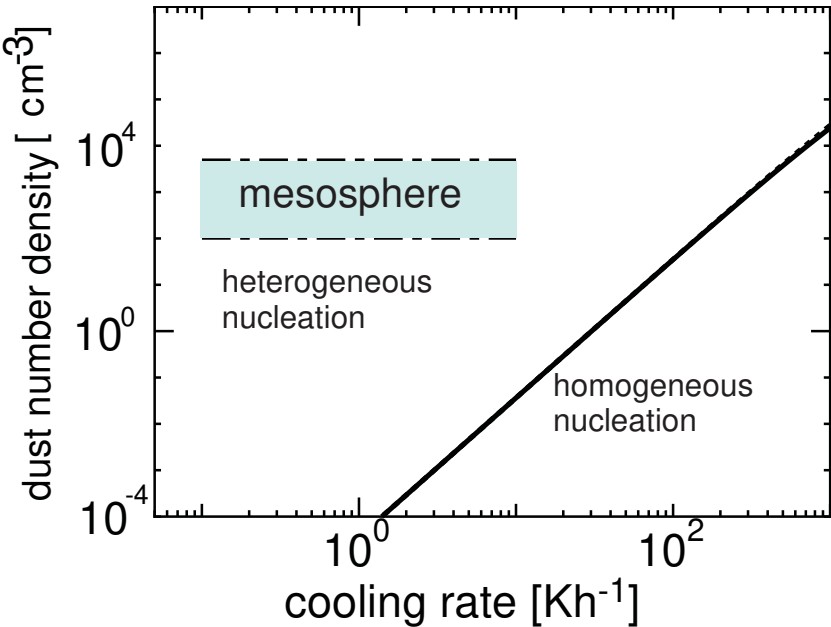

**Figure 7.** The same as Fig.6 but for an initial temperature of 145 K.

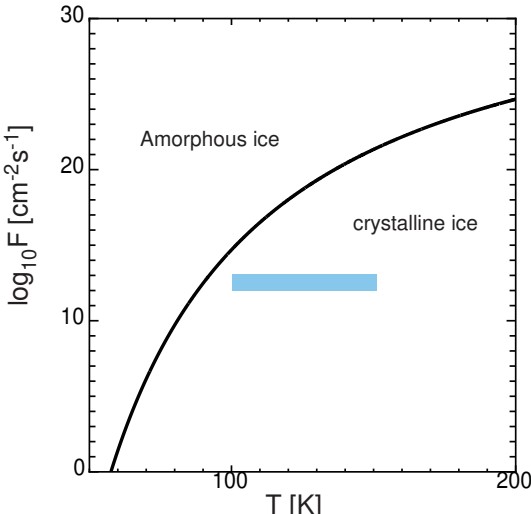

**Figure 8.** Flux of water molecules given by $F_c$ and shown by the solid line. In the region larger than $F_c$, the solid ice is considered to be amorphous, while in the smaller region it is crystalline. In the mesosphere, the flux of water molecules $F$ is approximately $10^{12} - 10^{13}$ cm$^{-2}$s$^{-1}$ for the temperature range of 100 K-150 K (shaded region).

s

condition:

$$F > D_s/a_1^4 = F_c, \tag{27}$$

where $F$ is the flux of water molecules and $D_s$ is the surface diffusion coefficient of the water molecules, which is given by $D_s = D_{s0}\exp(-E_s/kT)$. Figure 8 shows $F_c$ in Eq. (27) as a function of temperature, where $E_s/k = 4590$ K and $D_{s0} = 1.74 \times 10^5$ cm$^{-2}$s$^{-1}$ (Kouchi et al., 1994). In the region larger than $F_c$, the solid ice is considered to be amorphous. However, in the smaller region, the ice is crystalline. Figure 8 also shows the flux of water molecules on the dust surfaces $F$, which is assumed to be $F = n_1 v_{\rm th}$. In the mesosphere, the flux of water molecules is $\sim 10^{12} - 10^{13}$ cm$^{-2}$s$^{-1}$ (shaded region in Fig. 8). This flux range corresponds to crystalline ice formation. The results indicate that the ice particles solidify as crystals **when they condense through heterogenuous nucleation under mesospheric conditions.**

## 4  Discussion and conclusion

To explain the formation of clouds in the mesosphere, there are two possibilities: homogeneous and heterogeneous nucleations. We tested these two mechanisms theoretically. For homogeneous nucleation, we used the SP model, which agreed with the experiments and molecular dynamics simulations. The different nucleation models produce large differences in the nucleation process, mainly regarding the nucleation temperature. Using the nucleation rate obtained from the SP model, we calculated the time evolution of the number of water molecules and ice particle growth. Compared to the CNT model, the nucleation

temperature was very low. At an initial temperature of 135 K, the ice nucleation temperature was very low, ranging from 50 to 80 K (Fig. 4). When the initial temperature was 145 K, the number density of water molecules and the nucleation temperature both increased, but the nucleation temperature was still below (Fig. 5). The nucleation temperature for homogeneous nucleation is far below 100 K, and therefore, below typically observed temperatures. If the cooling rate was slower than $10^{-2}$ Kh$^{-1}$, then the nucleation temperature was above 100 K. However, the cooling time in the mesosphere is a few days at most; thus, the cooling rate will not be that slow. Therefore, the potential for homogeneous nucleation in the mesosphere is considered to be very small, although previous studies have suggested that homogeneous nucleation can occur.

We also determined the conditions at which heterogeneous nucleation occurs and compared them with observational data. Our results indicate that heterogeneous nucleation occurs effectively in the mesosphere. Because dust from micrometeorites is present at this altitude, heterogeneous nucleation using fragments of micrometeorites as nuclei is considered to occur significantly. As shown in **Section 3.3**, heterogeneous nucleation prevails even for a wide range of cooling rates and amounts of water in the mesosphere. When ice deposits due to heterogeneous nucleation, the growth rate is $(0.3 - 7) \times 10^{-3}$ nms$^{-1}$ from Eq. (19). This indicates that the radii of the particles increase to 1–25 nm in one hour. Since the clouds are observed on a timescale of a few hours, this rate is consistent with the observations. The particle growth rate becomes faster as the number density of water molecules increases; therefore, if rapid growth is observed, the number density of water molecules may need to be larger.

Our study also shows that during the deposition process, the ice can form directly in crystalline state rather than amorphous state. **The phase of ice particles in polar mesospheric clouds (PMCs) was determined using observations of the infrared extinction of the mesosphere from the Solar Occultation for Ice Experiment (SOFIE) on the AIM satellite (Hervig and Gordley, 2010). The observations could be explained using refractive indices of crystalline ice as opposed to amorphous ice; hence suggesting that not amorphous ice particles but rather particles of cubic ice existed near the mesopause (Hervig and Gordley, 2010). This observational result is consistent with our theoretical results that the nucleation leads to the formation of crystalline ice.**

**In this study, we obtained two different conditions for homogeneous nucleation. The first is the temperature that needs to prevail in the mesosphere so that homogeneous nucleation can occur. From this condition, we find that low cooling rates ( $\lesssim 10^{-2}$ K h$^{-1}$) are needed for the homogenous nucleation to be effective. These low cooling rates are unlikely in the mesosphere. The second condition is that homogeneous nucleation needs to be predominant in comparison to heterogeneous nucleation when dust grains are present. For this condition, a high cooling rate ( $\gtrsim 10$ K h$^{-1}$) is required. There is no overlap in the cooling rate value derived from these two conditions.** It is therefore unlikely that homogeneous nucleation is the major process for the formation of mesospheric cloud and noctilucent cloud particles. While homogeneous nucleation is unlikely to occur on Earth, the ice formation in the mesosphere is thought to be the most likely place on Earth for homogeneous nucleation to occur. Our results, however, **may** suggest that there is no particle formation via homogeneous nucleation on Earth. On the other hand, the probability for heterogeneous nucleation is very high even for small fraction of dust being present. After nucleation, the coagulation process for the formation of larger ice particles, which needs to be investigated based on different theories, should be studied in future research.

*Author contributions.* IM conceived the project, and IM and KT designed the study. KT performed the calculations and created the figures. YK contributed to the discussion of the paper. All authors contributed to writing and editing the manuscript.

*Competing interests.* The authors declare no conflicts of interest associated with this manuscript.

*Acknowledgements.* This work was supported in part by JSPS KAKENHI Grants No. 20H05657, 19K03941, and 18K03689 and by Re-
search Council of Norway Grant No. 275503.

355

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
