# Peer review of "Formation of ice particles through nucleation in the mesosphere"

_Atmospheric Chemistry and Physics, 2021_

## Author Comment (AC1)

*Response to Reviewer I*

*Reviewer's comments: The nucleation mechanism of polar mesospheric clouds has been a longstanding problem. At least since the late 1960s it has been debated whether homogeneous or heterogeneous nucleation is the dominant nucleation mechanism leading to the formation of ice particles under the extreme conditions of the polar summer mesopause (e.g., Witt 1969). While homogeneous nucleation had been deemed very unlikely given the contemporary knowledge on temperatures, water vapor mixing ratios and the (at that time only conceived) occurrence of meteoric smoke particles, observations of extreme gravity wave-induced temperature perturbations by Lübken et al. (2009) triggered Murray and Jensen (2010) to reinvestigate the problem. Based on (slightly modified) classical nucleation theory they concluded that homogeneous nucleation could indeed lead to the formation of amorphous solid water particles in the mesopause region if such extremely strong gravity wave-induced temperature perturbations (and hence cooling rates) occurred. However, they also found that if homogeneous nucleation had to compete with heterogeneous nucleation on meteoric smoke particles, the latter was more efficient and homogeneous nucleation became negligible.*

*In their current manuscript Tanaka and coauthors reconsider this problem based on the fact that the classical nucleation theory used in the work of Murray and Jensen is known to strongly disagree with laboratory observations for the case of water. Hence, Tanaka et al. apply a semi-phenomenological model which is known to be in much better agreement with observations. This model shows a much higher free energy barrier for nucleation such that homogeneous nucleation of ice particles in the mesopause region would require unrealistically low temperatures, i.e., well below 100K. Hence, this nucleation pathway can be ruled out (because it contradicts observed temperatures) while heterogenous nucleation is found to be feasible (in agreement with the recent groundbreaking laboratory measurements by Duft et al. 2019).In all this is a sound study that contributes to the important fundamental problem of ice nucleation in the mesopause region. While the study of Murray and Jensen predicted homogeneous nucleation to possibly occur under extreme, but still conceivable conditions (extreme cooling rates, no competing meteoric smoke) the work by Tanaka et al. now clarifies that even under such extreme conditions homogeneous nucleation cannot be expected. This result certainly warrants publication.*

*My recommendation is hence to publish this work provided that the following mostly minor issues are properly addressed before publication:*

*The referencing in the introduction could be improved by referring to the original papers for the statements made. Here are my suggestions:*

**Thank you for your valuable comments. We also appreciate many references that you provided. All the comments were for the improvement of the paper. We have added all references listed by the reviewer and revise our paper according to the reviewer's comments.**

*- line 16/17: original reference for noctilucent clouds: Jesse 1885; maybe also Vestine 1934*

**We have added two references to the paper.**

*- line 20: the original reference for satellite-based PMC observations is Donahue et al 1972; a very good review until 2006 is DeLand et al., 2006.*

**We have added the references.**

*- line 21: reference for particle sizes: Thomas and MacKay, 1985, von Cossart et al., 1999;*

**We have added the references.**

*- line 23/24: well, this statement is not correct as it stands here: the ground based visual sightings of NLC actually do not show a unique trend as shown in Kirkwood and Stebel (2003); however, a trend is observed in the brightness of satellite-based PMC observations as presented in Thomas et al (2003) and updated in DeLand and Thomas (2015).*

**Thank you for your useful comments. According to the reviewer's comment, we have added the references and rewritten the description.**

*- line 25: while the reference to Lübken et al. (2018) is good, the original paper posing this hypothesis should also be mentioned, i.e., Thomas et al., Nature 1989.*

**We added the reference.**

*- line 27: original reference on gravity wave-NLC-interaction: Witt, 1962.*

**We added the reference.**

*- line 28: to my knowledge temperatures as low as 100K (and even lower) have only been reported in Lübken et al. (2009). Lübken 1999 is a climatology for mean temperatures at 69°N (from falling sphere measurements); Rapp et al. (2002) do show gravity wave perturbed temperature measurements in NLC but with minimum temperatures of 110K.*

**We thank for the useful comment. As the reviewer pointed out, the value of the minimum temperature due to gravitational waves is 110 K, so we revised the text. We also cited Lübken et al. (2009) showing an observed value of 100K.**

*- line 33: homogeneous nucleation has only been considered feasible again (after many years during which it was regarded extremely unlikely) after Lübken et al. (2009) reported enormous temperature variability due to gravity waves (see their figures 9, 10 and 11). Until then the consensus in the community was that it was rather heterogenous nucleation on meteoric smoke (see e.g., Rapp and Thomas 2006 for a discussion).*

**Thank you for your comments on the consensus of the community so far. It is an important point, so we include the statement in our paper.**

*- line 39 and 40: The authors are mixing two things here: as reviewed in Rapp and Thomas, the stated species have been suggested in the literature as potential nuclei for mesospheric ice particle formation. However, not all of the stated species are candidates for the composition of meteoric smoke (e.g., proton hydrates, soot are independent of meteoric origin). Meteoric smoke composition is indeed discussed in Plane (2015). I recommend to have a look at this paper and change the sentence accordingly.*

**Thank you for pointing this out. According to the comment, the description of meteor smoke was not accurate, so we revised the description.**

*- Section 3: in order to put the results in perspective, it would be useful if the authors included a short section describing typical ranges of mesospheric variables like observed temperatures, water vapor mixing ratios or partial pressures, concentrations of meteoric smoke particles (e.g. from rocket borne observations), and cooling rates due to tides and gravity waves. This will help assessing the assumptions made and results achieved in the paper. In this context, the authors should clearly state if derived or used values are way outside of observed ranges.*

**Accoding to the reviewer's comment, we included a short subsection describing typical ranges of mesospheric variables in Section 3.1.**

*- line 238/239: the authors should point out that cooling rates as low as $10^{-6} Ks^{-1}$ at initial temperature of 135K also corresponds to a completely unrealistic time that the nucleation would take. However, observations do show that PMSE (which are also evidence for ice particles, but already at times when they have not yet grown large enough to be optically detectable) form rapidly for example in updrafts of gravity waves (i.e., within minutes).*

**As the reviewer pointed out, the cooling rate of $10^{-6}$ Ks$^{-1}$ is unrealistic in the mesosphere. So we added the description in Section 3.**

*- line 235/236: These formulations are misleading. "the amount of water vapor present was 20 times higher at 145K than at 135K" – this certainly doesn't have anything to do with the atmosphere. In the atmosphere, the water vapor mixing ratio in the mesosphere is determined by transport across the tropical tropopause and oxidation of methane in the stratosphere (roughly at a ratio 50:50) and does not depend on the local temperature. Please clarify what you mean.*

**As the reviewer suggested, it was a misleading expression, so we rewrote it. We meant that for the equilibrium vapor pressure, the number density at 145 K is 20 times greater than 135 K. We have changed it to such an expression. On the other hand, as the reviewer pointed out, the number density of water molecules depends on various other factors, so we included the description in Section 3.1.**

*- Figures 6 and 7: please give "dust density" in number densities and not mass densities for easier interpretation in terms of known values from previous models and observations.*

**According to the reviewer's suggestion, we rewrote the vertical axis of the graph as the number density. We have also revised the derivation of the conditional equation in the text to reflect this change in Section 2.**

*- Section 3.3: these are important results. The authors should maybe also state that measurements with SOFIE on AIM can only then be properly explained if the refractive index for crystalline ice is used, but not for amorphous ice. I remember that this was presented by Mark Hervig at several meetings. The authors might like to check back with him where this is published.*

**Thank you for your useful comment. We contacted him and cited the reference in our paper. We also added the description about the measurements with SOFIE in Section 4:**

**The phase of ice particles in polar mesospheric clouds (PMCs) was determined using observations of the infrared extinction of the mesosphere from the Solar Occultation for Ice Experiment (SOFIE) on the AIM satellite (Hervig and Gordley, 2010). The observations could be explained using refractive indices of crystalline ice as opposed to amorphous ice; hence suggesting that not amorphous ice particles but rather particles of cubic ice existed near the mesopause (Hervig and Gordley, 2010). This observational result is consistent with our theoretical results that the nucleation leads to the formation of crystalline ice.**

---

## Author Comment (AC2)

Reply to Reviewer II

*1. This manuscript addresses the microphysics of noctilucent cloud formation in the mesosphere. The authors make an important case that a semi-phenomenological model is better suited for describing ice particle nucleation under mesospheric conditions than classical (or modified classical) nucleation theory. They apply the model to investigate whether homogeneous or heterogeneous nucleation dominates the formation of noctilucent clouds. The idea is good of directly comparing the nucleation rates resulting from homogeneous and heterogeneous nucleation under given mesospheric conditions. The conclusion that heterogeneous nucleation is expected to dominate under most conditions is reasonable and in line with earlier studies.*

*A major problem of the approach is the nucleation scenario that the authors adopt. They assume a process of continuous temperature decrease down to a very low temperature at that the nucleation rate reaches a maximum. They call this temperature "nucleation temperature" and use it as a characteristic parameter describing the cloud nucleation process. I argue that this scenario is not relevant for mesospheric conditions. Sufficient homogeneous nucleation can occur at temperatures substantially above the authors' "nucleation temperature". In fact, it would be very much counter-productive for the formation of noctilucent clouds if these (unrealistically) low "nucleation temperatures" were reached in the mesosphere.*

**We thank for the reviewer's valuable comments. As the reviewer pointed out, we defined the nucleation temperature in our paper. Although our method is a simplification, we believe that it is applicable to mesospheric conditions. The reviewer pointed out that temperatures higher than the nucleation temperature would be sufficient for homogeneous nucleation to occur, but we think this is difficult, because the nucleation rate is very low in the temperature range from the nucleation temperature to around 100K. In the mesosphere, the minimum temperature is never lower than 100K, so we concluded that the homogeneous nucleation is very unlikely to occur.**
**The explanation was inadequate, so we added the description in the revised paper, as stated in the reply to the comment 2.**

2. *This can be illustrated using Figure 2. Here, the semi-phenomenological model has been used to describe a very slow cooling process, starting out from a typical polar summer mesopause temperature (135 K) and then extending over more than 30 hours. After this time, a homogeneous nucleation rate of about 1 $cm^{-3}$ $s^{-1}$ is reached at a "nucleation temperature" Tp of about 65 K. However, it is not necessary to reach this maximum nucleation rate in order to form noctilucent clouds. Already a nucleation rate of e.g. 0.01 $cm^{-3}$ $s^{-1}$ leads to an ice particle concentration of about 100 $cm^{-3}$ after few hours. This is sufficient for noctilucent clouds, and it is achieved at significantly higher temperature. Moreover, at 65 K the nucleation of rate 1 $cm^{-3}$ $s^{-1}$ will lead to so many nucleation events that competition for the available water vapour will prevent the individual particles from growing large. This will make it impossible to form visible noctilucent clouds (that typically require particle radii exceeding 20 nm). I thus argue that Tp is not a meaningful parameter to describe homogeneous nucleation of noctilucent clouds in the mesosphere. It follows that cooling rates slower than 1e-5 K $s^{-1}$ are not a requirement for the occurrence of homogeneous nucleation in the mesosphere (lines 238-239). Also, the very strong statement "there is no particle formation via homogeneous nucleation on Earth" in the Conclusions (lines 295-296) does not hold based on the Tp analysis.*

**Figure 2 shows the nucleation rate in the cooling process. The reviewer pointed out that the nucleation temperature does not have to be 65 K, and that a larger nucleation rate of 0.01 $cm^{-3}$ $s^{-1}$ is sufficient. However, the temperature, at which such a large value as 0.01$cm^{-3}$ $s^{-1}$ is realized, is very low less than 100K. As shown in Figure 2, for example, the nucleation rate is only $10^{-5}$ $cm^{-3}$ $s^{-1}$ at 85K. This indicates that only about 1 $cm^{-3}$ of nuclei is formed in one day. Considering that the temperature in the mesosphere is larger than 100 K, we concluded that the homogeneous nucleation is difficult.**

**We showed the nucleation temperature at which the nucleation rate peaks in the cooling process, but did not the monomer consumption. The temperature at which the nucleation rate peaks and the temperature at which the number density decreases steeply are very near, so we consider Tp is meaningful. Since Figure 2 did not show the variation of monomer consumption due to condensation, we added the number density change of vapor in Figure 2. The wording has been slightly changed for the strong statement in the conclusions pointed out by the reviewers.**

[Figure]

**Fig.2 Time evolution of the nucleation rate and mean particle radius during homogeneous nucleation (upper panel), and the ratio of the number density of water molecule to the initial value (bottom panel) calculated using the SP model. The initial temperature was 135 K and the cooling rate is 3.6 K h$^{-1}$.**

3. *There are more inconsistent statements about the cooling rates. In section 3.1 it is argued that very slow cooling rates (< 1e-5 K s-1) are necessary for homogeneous nucleation (lines 238-239). In section 3.2, on the other hand, it is concluded that high cooling rates (> 1e-2 K s-1) are needed for homogeneous nucleation to be important (figures 6 and 7). This contradiction needs to be discussed.*

We are investigating two different conditions for homogeneous nucleation. The first condition is not a requirement for homogeneous nucleation to occur, but a limitation on the temperature, that is, the condition required for condensation temperature realized in mesosphere when homogeneous nucleation occurs. From this condition, we found that a slow cooling rate (< $10^{-5}$ K s$^{-1}$) is required. The second condition is that the

homogeneous nucleation is dominant in the presence of dusts. For this condition, a high cooling rate (> $10^{-2}$ K s$^{-1}$) is required. The fact that the two conditions do not have an overlapped region of the cooling rate indicates that the homogeneous nucleation is difficult to achieve. We should include this description, so we included it in the revised paper.

4. *Cooling rates in the manuscript are expressed in units K s$^{-1}$ (e.g. figures 4 and 5). Using the unit K h$^{-1}$ would be much more instructive and would provide the reader with a better feeling for the mesospheric processes. The authors should discuss what cooling rates can typically be expected in the mesosphere. One could e.g. consider the cooling rate connected to a typical gravity wave of amplitude 10 K and period of a few hours.*

According to the reviewer's comment, we changed the unit (K h$^{-1}$) for all descriptions and graphs in the revised paper. Typical values of the cooling rate obtained from previous studies have been added in the new section 3.1.

*5. Some comments concerning the heterogeneous nucleations:*

*- Line 157-158: It is stated "As indicated above, the radius of the critical cluster is very small*

*(i = 2−10), making this assumption reasonable." I do not find where "above" this is indicated. Please provide a justification why the critical radius is so small (i = 2-10).*

The calculation of homogeneous nucleation gives information about the size of critical nuclei. As mentioned in the paper, the sizes of the critical clusters are very small in both models, e.g., two and four molecules for the SP and MCNT models for the case of Fig.3. The cause of small critical cluster is due to extremely high supersaturation ratios: From the nucleation theory, the size of critical cluster, $i_*$, is given by $\left(\frac{2\eta}{3lnS}\right)^3$ for the CNT

and MCNT, and $i_* = \left(\frac{\eta+\sqrt{2\eta+3\xi lnS}}{3lnS}\right)^3$ for the SP model. In our calculation ranges, the size of critical nuclei ranged from 2 to 10. As the reviewer pointed out, the description was insufficient. We have therefore added the description of how the critical nucleus is determined in Section 2 and explained the size of critical cluster in more detail.

*- Line 180-182: The condition given by equation 18 states that at least 50% of the initial molecules n$_1$(0) in the water vapour are consumed by heterogeneous ice particle growth. In*

*contrast to this, lines 181-182 state that the number density of water vapour is largely unchanged during the nucleation process, i.e. water vapour is largely not consumed. The latter statement is the basis for the linear growth of the ice partile radus with time described by equation 19. This seems to be a contradiction that would make equations 20-24 invalid.*

**As the reviewers pointed out, the estimates were rough. So we corrected the condition within a reasonable range and changed the threshold from 50% to 10%. The figure and description were revised to reflect this change. With this condition, we obtain the number density of dust particles for the significant start of the heterogeneous nucleation, rather than the predominance. We have rewritten the expression in the paper. Although our estimates are rough, we consider them useful because it is a straightforward formulation of how the number density of smoke dusts necessary for the heterogeneous nucleation depends on the dust size and water vapor content.**

*6. Some comments concerning the equations:*

*- equations 12 and 13: r1 in these equations should be r0 in order to be consistent with the radius of the monomer defined in line 103.*

**We corrected it. The monomer radius was unified to be $r_1$ in the revised paper.**

*- equation 18: "r" should be replaced by "a", the radius of the dust grain.*

**We corrected it.**

*Some comments concerning the Introduction:*

*- Line 15: "mesosphere" should be replaced by "mesopause region".*

**We corrected it.**

*- Line 19: The sentence "Ice particles, also known as polar mesospheric clouds, have recently been observed by satellites (Hervig et al., 2012)" should be rephrased. "Noctilucent clouds" are also known as "polar mesospheric clouds". Polar mesospheric clouds have been observed by satellites not only "recently" but as early as in the 1970s.*

**We corrected it.**

*- Line 23: The authors seem to imply that noctilucent clouds "were considered to exist" before their discovery by observations. This is not the case.*

**As the reviewer said, this sentence was not explanatory enough and did not need to be between the preceding and following sentences, so we removed it.**

*- Line 23: What is meant by "[noctilucent clouds] were difficult to observe visually before the twentieth century"?*

**As mentioned in the above comment, we corrected it.**

*- Line 39: It is stated "Meteoric smoke particles consist of sodium bicarbonate, sodium hydroxide, soot, sulfuric acid, and proton hydrates". Soot, sulfuric acid and proton hydrates are indeed considered to be part of the middle atmospheric aerosol. However, they are not expected to be ingredients of meteoric smoke particles in the mesosphere.*

**According to the comment, the description of meteor smoke was not accurate, so we revised the description.**

*- Line 49: Please make clear that by "solid particles" in this sentence you mean meteoric smoke particles, not ice particles.*

**We corrected it.**

*- Line 73: Avoid the term "measured" when referring to molecular dynamics simulations. More suitable terms may be "studied" or "derived".*

**We changed the word.**

*7. Some editorial comments:*

*- Line 287: "Section 4" should be "Section 3.2".*

**We corrected it. We revised to Section3.3, since Section 3.1 has been added in the revised paper.**

*- Line 331: Please provide a complete reference.*

**We corrected it.**

*- Line 332: "Merner" should be "Megner".*

**We corrected it.**

*- Line 350: "Hoffner" should be "Höffner". "Rottger" should be "Röttger".*

 **We corrected it.**